# Effects of Oxiris^®^ Therapy on Cytokine Elimination after a LPS Infusion—An Experimental Animal Study

**DOI:** 10.3390/ijms25179283

**Published:** 2024-08-27

**Authors:** Armin Kalenka, Philipp Arens, Ralf M. Müllenbach, Markus A. Weigand, Maik Brune, Mascha O. Fiedler-Kalenka

**Affiliations:** 1Department of Anesthesiology, Medical Faculty, University Hospital Heidelberg, University Heidelberg, Im Neuenheimer Feld 420, 69120 Heidelberg, Germany; armin.kalenka@kkh-bergstrasse.de (A.K.);; 2Hospital Bergstrasse, 64646 Heppenheim, Germany; 3Department of Anesthesiology, Critical Care Medicine, Emergency Medicine and Pain Therapy, ECMO-Center, Campus Kassel of the University of Southampton, 34125 Kassel, Germany; 4Department of Internal Medicine I and Clinical Chemistry, Medical Faculty, University Hospital Heidelberg, University Heidelberg, 69120 Heidelberg, Germany

**Keywords:** inflammation, sepsis, hemadsorption, lipopolysaccharide, interleukin

## Abstract

The clinical effectiveness of Oxiris^®^, particularly in reducing cytokines, remains uncertain due to the limited data provided. This study explored and analyzed the application value of Oxiris^®^ endotoxin adsorption technology in a large animal model. Pigs received an intravenous LPS infusion. Six animals were treated 2 h after the infusion with an Oxiris^®^ hemadsorption using a pumpless extracorporeal technique for 6 h. Five animals served as controls. Cardiocirculatory parameters, hyperspectral analysis, and a panel of cytokines were measured. The lipopolysaccharide infusion induced sepsis-like inflammation with tachycardia, elevated pulmonary pressure, elevated lactate level, as well as elevated pro-inflammatory cytokines like interferon (IFN)-γ, interleukin (IL)-1β, IL-2, IL-6, IL-8, IL-12 and tumor necrosis factor alpha (TNF-α). In addition, increases of anti-inflammatory cytokines like IL-1ra and IL-10 were found. After 3 and 6 h in both groups, pro-inflammatory cytokines were significantly reduced. No differences between the intervention and the control group could be detected after 3 and 6 h for IL-1β, IL-2, IL-6, IL-8, IL-12 and TNF-α, suggesting no effect of the Oxiris^®^ filter on the elimination of elevated cytokines with a pumpless extracorporeal hemadsorption technique. The presented large animal model may be a promising option for studying the effects of hemadsorption techniques.

## 1. Introduction

Sepsis remains one of the main mortality causes in critically ill patients [1]. Sepsis is a life-threatening condition, most commonly triggered by bacterial and other infections, characterized by the host’s dysregulated inflammatory response, which involves both proinflammatory cytokines and a compensatory anti-inflammatory response [2]. Lipopolysaccharide (LPS), also known as endotoxin, is a well-documented component of the cell wall in gram-negative bacteria. For instance, endotoxin can initiate cellular biosynthesis, activate intracellular apoptosis mechanisms and strongly stimulate inflammatory pathways. This leads to the release of pro-inflammatory cytokines and chemokines such as tumor necrosis factor-alpha (TNF-α), interleukin-6 (IL-6) and interleukin-18 (IL-18), along with other bioactive metabolites associated with organ damage and septic shock [3,4].

Blood purification therapies are undergoing significant advancements and gaining traction in sepsis management, propelled by both technological enhancements and the steadfast belief among a segment of the medical community in their efficacy for septic shock [5]. The purification of molecules responsible for the disproportionate and/or abnormal response of the host to infection, such as endotoxins, pathogen-associated molecular patterns (PAMPs), damage-associated molecular patterns (DAMPs) and cytokines, is the focus of interest here [6].

In more recent developments, the Oxiris^®^ filter, incorporating three layers for adsorption, along with hemofiltration and antithrombotic priming [7], has demonstrated effective adsorption of LPS and cytokines in vitro [8]. Additionally, it has shown promise in reversing hemodynamic instability in animal models of septic shock [9]. However, the clinical effectiveness of Oxiris^®^, especially in achieving the desired decrease in cytokines, remains unclear [10].

Hemadsorption devices are typically incorporated into an extracorporeal circuit like continuous renal replacement therapy (CRRT) or extracorporeal membrane oxygenation (ECMO). Consequently, the clinical utilization of hemadsorption is often deferred until organ failure manifests, such as acute renal or lung failure, necessitating CRRT or ECMO therapy. A new pumpless extracorporeal hemadsorption technique (pEHAT) has recently emerged, offering an independent approach to hemadsorption that does not require an additional medical device [11]. We investigated cytokine elimination, as well as micro- and macrocirculatory parameters, in an animal study up to 6 h after a LPS challenge. 

## 2. Results

A total of 12 pigs (49 (37; 54) kg bodyweight) were included in the study. One pig died after the induction of anesthesia. The administration of LPS led to the expected inflammatory responses: tachycardia, elevated PAP, elevated lactate level, as well as elevated pro-inflammatory cytokines like interferon (IFN)-γ, interleukin (IL)-1β, IL-2, IL-6, IL-8, IL-12 and tumor necrosis factor alpha (TNF α). We also found significant increases of anti-inflammatory cytokines such as IL-1ra and IL-10. We observed elevated VO_2_ and VCO_2_ as well as a reduction in tissue oxygen (StO_2_) (Figure 1). MAP and CI remained stable after LPS infusion (Table 1).

Over time, between H2, H5 and H8, values for HR, PAP and StO_2_ remained unchanged in both groups, suggesting a persistent inflammatory response (Table 2). Lactate normalized in the control group and was still above 2 mmol/l in the Oxiris^®^ group (Figure 2). VO_2_ and VCO_2_ decreased in the Oxiris^®^ group but not in the control group (Table 2).

Pro-inflammatory cytokines such as IL-1β, IL-8 and TNF-α decreased significantly in both groups over time points 3 and 6 h after LPS. IL-6 was lower at H5 only in the Oxiris^®^ group. At H8 in both groups, IL-6 was lower than at timepoint H0 (Figure 3). At H5, no differences were detected for IL-1β, IL-6, IL-8, IL-10 and TNF-α between the groups (Table 3). No differences were found in pro-inflammatory cytokines at H8 between the two groups. This suggests no measurable effect of the Oxiris^®^ filter. Over time, anti-inflammatory cytokine IL-1ra levels were unchanged, and IL-10 decreased in both groups (Table 3). Interestingly, anti-inflammatory cytokine IL-1ra levels were lower in the Oxiris^®^ group at H8 compared to the control group (Figure 4).

## 3. Discussion

We selected a large animal pig model because porcine anatomy and physiology are generally quite similar to those of humans [12], making the transfer of knowledge more applicable. Specifically, the blood flow through the hemadsorption device closely resembles that in humans [11]. For a more detailed insight into inflammatory large animal models, we refer to a recent review [13].

Experimental endotoxemia primarily represents a systemic challenge without an initial infectious focus and lacks the characteristic immune response induced by sepsis [14]. We chose an established model for the controlled simulation of an inflammatory stimulus in pigs [15]. Due to its potency as an immune system stimulator, lipopolysaccharide has been extensively used across various animal species to investigate gram-negative bacterial infections and sepsis. LPS injection is widely accepted as a model for studying the hypodynamic phase of gram-negative bacterial shock rather than sepsis [16]. We opted for a high single bolus dose of LPS (100 µg/kg bw). Usually, doses ranging from 0.5 to 75 µg/kg bw are used [15]. We selected this dose to generate a relevant IL-6 response (4 pg/mL) as the primary marker of inflammatory response to the LPS stimulus [17,18,19]. However, using the high dose allowed us to achieve the desired cardio-pulmonary and inflammatory effect. Nevertheless, the administration of vasopressors was not necessary at any time. Higher LPS bolus doses up to 150 µg/kg bw produce an experimental model for endotoxemic shock in pigs [20].

We selected the time point 2 h after the LPS administration for randomization. This time point was selected based on previous studies describing the plasma concentration of pro-inflammatory cytokines after intravenous LPS challenge in pigs. The maximum IL-6 value was found between 2 and 4 h after a single dose of LPS [19]. TNF-α also peaks 1 to 3 h after administration [21], as does IL-1β [22].

Numerous other studies on pigs involving LPS investigate the prolonged administration of LPS [23]. The Oxiris^®^ filter, with its particular structure, allows to reduce cytokine levels and LPS by adsorption [23]. In vitro and human studies have shown that LPS adsorption reduces cytokines, systemic inflammation and vasoplegia [24]. Therefore, we did not use a prolonged LPS infusion and started the Oxiris^®^ filter 2 h after the LPS infusion.

In our model, pro-inflammatory cytokines such as IL-1β, IL-6, IL-8, and TNF α, as well as anti-inflammatory cytokines such as IL-1ra and IL-10, were significantly higher two hours post LPS infusion. Sepsis is characterized by impaired cellular oxygen utilization despite adequate oxygen delivery, with cytopathic hypoxia recognized as a mechanism of resulting organ dysfunction [25]. On our model, we measured elevated VO_2_ and VCO_2_ levels after LPS infusion. Interestingly, these elevations persisted in the control group after 8 h post LPS administration, whereas this was not the case in the Oxiris^®^ group. Elevated VO_2_ and VCO_2_ are characteristic in septic patients [26]. In a recent study on critical ill COVID-19 patients, the association of S_t_O_2_ and the severity of organ dysfunction were described [27].

In summary, after the LPS infusion in our model, we were able to identify relevant changes in the cardio-pulmonary and inflammatory response indicative of a septic condition.

Studies have confirmed that elevated endotoxin levels in the body can significantly increase patient mortality [28]. As a result, the prompt and effective removal of endotoxins and inflammatory mediators may be a crucial treatment option. The current evidence supporting the use of extracorporeal hemadsorption techniques is not conclusive. However, available data indicate that these techniques are safe and most effective during the early phase of septic shock, particularly within the first 24 h of the dysregulated host response [29,30]. A human study involving 24 volunteers who were challenged with a LPS infusion showed that, in subjects of the extracorporeal hemadsorption (CytoSorb^®^) group, the increase in plasma cytokine concentrations after the first LPS challenge was significantly attenuated compared to the control group. In this study, hemadsorption was initiated as early as 15 min after the initial LPS bolus [31]. In an animal study in a porcine model of smoke and burn injury, hemadsorption with the CytoSorb^®^ system did not lead to significant reductions in the measured cytokines [32].

The Oxiris^®^ membrane material is an advanced iteration of the AN69 hydrogel structure. The core material, AN69 (a polymer of propylene and sodium methanesulfonate), can adsorb various pro-inflammatory and anti-inflammatory cytokines and molecules. Its surface is modified with a polyethyleneimine cationic polymer, giving it a multilayer linear structure and a positive charge. This allows it to filter and remove blood, adsorbing negatively charged endotoxins through ionic bonding [8,24]. As a result, it may offer the most comprehensive functionality among current blood purification products [33]. A recent meta-analysis of 14 studies, 10 observational and four randomized control trials, all conducted at single centers, revealed limited data regarding the effectiveness of the Oxiris^®^ filter [34].

A human study showed that when Oxiris^®^ was implemented in a RRT system, the average blood flow rate was 132 ± 69 mL/min [24]. A cross-over regime consisting of 24 h treatment with an Oxiris^®^ filter followed by 24 h treatment with a standard filter or the reverse filter order was used in this study. Cytokine levels significantly decreased over time in both groups. TNF-α decreased more significantly with the Oxiris^®^ filter, while IL-6 and IL-8 remained at similar levels in both groups.

In a non-randomized study, RRT was conducted using either the Oxiris^®^ or a classic AN69-ST filter with a blood flow rate between 150 and 200 mL/min. The choice of filter was determined by the patient or their relatives, since the Oxiris^®^ filter was not covered by insurance, and it was not the preferred option for some patients. In this study several cytokines decreased in both filters after 24 h, significantly more so in the Oxiris^®^ group [35].

A retrospective observational study included 70 patients with suspected or proven gram-negative infection and acute renal failure. These patients were treated with Oxiris^®^ filter and were matched to 66 historical patients as controls. IL-6 levels were higher at the start of the study in the Oxiris^®^ group and decreased in both groups, with a greater reduction observed in the Oxiris^®^ group [36].

Twenty-three septic patients were treated with Oxiris^®^ in another study. The analyses included another 30 patients who were treated at the same time period, serving as controls. The levels of IL-6, IL-10 and endotoxin in both groups decreased significantly. Moreover, the Oxiris^®^ group showed significantly lower levels of these cytokines compared to the control group after 3 days of treatment [37].

The abovementioned four studies introduced the Oxiris^®^ filter together with an RRT. The “peak concentration hypothesis” proposes that continuous RRT could be advantageous for the treatment of sepsis. Due to their continuous nature and non-specific removal capabilities, RRT may help reduce the peaks of both pro-inflammatory and anti-inflammatory mediator concentrations, thereby restoring immunohomeostasis [38]. Several studies have indicated that cytokine removal by RRT is primarily attributed to adsorption rather than convective transport [39,40].

We recently introduced a new pumpless extracorporeal hemadsorption technique (pEHAT), which functions as an HA method without the need for an additional medical device [11]. Therefore, no convective elimination, as occurs during continuous RRT, takes place, making it a “pure” hemadsorption technique. Blood flow over the arteriovenous shunt with a MAP above 75 mmHg, which was consistently the case in the recent study, is above 300 mL/min [11]. Therefore, we generated a blood flow higher than with a classical RRT.

In the actual study, we began the treatment with Oxiris^®^ 2 h after LPS infusion. The elevated pro-inflammatory cytokines (IL-1β, IL-6, IL-8) decreased in both groups, as well as the anti-inflammatory IL-10. At H5 and H8 (after 3 and 6 h treatment in the Oxiris^®^ group), no significant difference could be found for pro-inflammatory cytokines and IL-10 between the groups. Interestingly, lactate in the control group decreased between H2 and H5, as well as between H2 and H8, which was not the case in the Oxiris^®^ group. Nevertheless, we could not observe a significant difference between the two groups at H8. In addition, we could not find any significant differences in our model of “pure” absorption technique at H5 and H8 between the groups for the following parameters: HR, PAP, lactate, StO_2_, VO_2_ and VCO_2_.

IL-1ra, known as a natural anti-inflammatory cytokine, was lower in the Oxiris^®^ group compared to the control group at H5 and H8. This phenomenon has already been described in septic patients undergoing high-volume hemofiltration with a blood flow between 250 and 350 mL/min [41]. The therapeutic use of Anakinra, a non-glycosylated human IL-1ra, in septic shock patients is still under debate [42,43]. However, the high, non-selective adsorption of an anti-inflammatory cytokine may not be meaningful.

Our study contains several advantages. We utilized a uniform stimulus to induce inflammation in a simulated, large animal model. We conducted extensive analyses at both micro- and macrocirculatory levels, as well as across a broad panel of cytokine values. Using the pEHAT method, we eliminated convective elimination associated with traditional RRT.

This study has several limitations. First, this is a large animal model and therefore the results cannot be applied to human patients without any restrictions. Secondly, the relatively small number of animals per group is probably not sufficient for the analysis results to be considered relatively reliable. Third, we simulated a gram-negative infection through LPS infusion. The effects of HA in a gram-positive infection may be different. Fourth, we do not have a comparison group with the Oxiris^®^ filter using the pEHAT technique and a classical RRT.

In summary, an infusion of 100 µg/kg in this large animal model resulted in a significant inflammatory response. Both pro-inflammatory and anti-inflammatory cytokines in the plasma increased significantly following LPS administration. We implemented a pumpless extracorporeal hemadsorption technique (pEHAT), which functions as an HA method, in the intervention group for 6 h without detectable problems. However, we were unable to demonstrate any significant effects of the Oxiris^®^ filter on the elevated cytokines following LPS infusion in this large animal model. We aimed to present a clear data analysis within the context of existing evidence in our paper. From our perspective, our data support the use of hemoadsorption only within the framework of a clinical study. In our opinion, these studies should include coordinated patient selection and measurements of cytokine levels. The presented large animal model may be a promising option for studying the effects of hemadsorption techniques.

## 4. Materials and Methods

### 4.1. Animal Preparation

The study received approval from the responsible animal research committee (governmental district authority Karlsruhe 35-9185.81/G-109/20). After an overnight fast with free access to water, female domestic pigs were anaesthetized intramuscularly using a combination of 7 mg/kg Azaperon (Stresnil^®^, Lilly, Bad Homburg, Germany), 2.5 mg/kg S-Ketamin (Pfizer Pharma, Berlin, Germany) and 0.3 mg/kg Midazolam (Midazolam, Hameln Pharma, Hameln, Germany). The pigs were then intubated and mechanically ventilated with a Carescape R860^®^ ventilator (GE Healthcare, Madison, WI, USA) using an inspiratory oxygen concentration (F_i_O_2_) of 0.4 in volume-controlled mode. The ventilation parameters included a tidal volume of 8 mL/kg body weight (bw), an inspiration/expiration ratio of 1:2 and a PEEP of 5 cm H_2_O. Consumption of oxygen (VO_2_) and carbon dioxide (VCO_2_) was recorded with the ventilator. Anesthesia was maintained through the continuous infusion of 6 mg/kg/h S-Ketamin and 3.6 mg/kg/h Midazolam. The depth of anesthesia was regularly assessed by ensuring the absence of spontaneous breathing efforts.

### 4.2. Instrumentation and Measurements

All potential puncture sites were thoroughly washed with soap. Invasive catheters were placed following skin disinfection and under sterile conditions. An arterial catheter was inserted with ultrasound guidance (VScan^®^, GE Ultrasound, Horten, Norway) into the left internal carotid artery to monitor the systemic arterial pressure and to determine arterial blood gas samples. A pulmonary arterial catheter was inserted via the right external jugular vein for pulmonary arterial pressure (PAP) and continuous cardiac output measurements (Vigilance II^®^, Edwards Lifesciences, Irvine, CA, USA).

After the initial instrumentation, baseline measurements (heart rate (HR), mean arterial pressure (MAP), pulmonary artery pressure (PAP), cardiac index (CI), blood gases, VO_2_ and VCO_2_ and hyperspectral imaging were measured. An intravenous dose of LPS (100 µg/kg bw O111:B4, LPS-EB, InvivoGen, Toulouse, France) was then administered over 15 min. Measurements were recorded immediately after LPS administration and every hour following the baseline measurements. For the sake of simplicity these data are presented in the tables at baseline (H0), 2 h (H2), 5 h (H5) and 8 h (H8) only. Cytokine levels were collected at baseline (H0), 2 h (H2), 5 h (H5) and 8 h (H8). The animals were assigned to the two groups using a computer-generated randomizer: Group A as the control group without pEHAT, and Group B with pEHAT treatment starting 120 min after LPS infusion (H2) and monitored for a further 6 h (Figure 5).

In Group B, an arteriovenous shunt was established between the femoral artery and femoral vein. A small 10 French arterial cannula (Bio-Medicus^®^, Medtronic, Minneapolis, MN, USA) was inserted under steril conditions as the arterial line, and a 14 French venous cannula (Bio-Medicus^®^ Medtronic, Minneapolis, MN, USA) was utilized for venous return to allow the most unresisted outflow possible [11]. Anticoagulation was achieved by continuous intravenous heparin. 

### 4.3. Hyperspectral Imaging (HSI) Measurements

HSI uses the specific light reflection and absorption of substances such as oxy-/deoxyhemoglobin or water to measure tissue oxygenation, perfusion and the water content of tissue. For the HSI measurements, the TIVITA^®^ Tissue System (Diaspective Vision GmbH, Am Salzhaff, Germany) was applied. It offers a non-invasive evaluation of three parameters: tissue oxygenation (S_t_O_2_), near infrared perfusion index (NIR) and tissue water index (TWI). The S_t_O_2_ scales the tissue oxygen saturation (0–100%) on the surface with a penetration depth of up to 1 mm. NIR measures the oxygen saturation and perfusion of deeper tissue layers (4–6 mm penetration depth). NIR and TWI are index values (0–100) indicated in predefined arbitrary units. The three HSI-parameters are displayed as color-coded images. Red/yellow areas indicate high values (50–100) and green/blue areas indicate low values (0–50), and HSI measurements were performed on the ear of the swine [44].

### 4.4. Cytokine Analysis

Plasma was isolated from whole blood via centrifugation and stored at −20 °C for further use. A Luminex-based multiplex assay (Millipore porcine cytokine/chemokine 13-Plex, Merck, Darmstadt, Germany) was used according to the manufacturer’s instructions. Samples were run in triplicate against internal controls, and the mean value was used for further statistical analysis.

We estimated an elimination half-time for IL-6 and TNF-α of three hours based on available cytokine kinetics from porcine models [45,46].

The entire experimental setting took eight to nine hours. At the end of the experimental protocol, the pigs were euthanized.

### 4.5. Data Analysis

Baseline values were tested for normal distribution. In the case of a normal distribution, a paired t-test was performed; otherwise, the Wilcoxon test was used to compare values between the group at timepoint H0 and H2 (SPSS, IBM, Version 28.0). Data between groups were analyzed by Mann–Whitney U test. Values are shown as median (minimum; maximum) values. A *p*-value of less than 5% (*p* < 0.05) was considered statistically significant.

Due to the preliminary nature of the study, we chose a group size of 6 animals per group. Nonetheless, we based our calculations on a peak IL-6 concentration of 4 pg/mL. With an expected native half-life elimination of 50% (2 pg/mL) and an increased doubled elimination due to the filter, we conducted a power calculation that resulted in a minimum sample size of 5 animals per group (control group: 2 ± 0.5 pg/mL, Oxiris^®^ Group: 1.0 ± 0.5, α = 0.05). Sex had a significant impact on the immune responses after the LPS challenge. We therefore used only female pigs, which exhibit greater TNF-α and IL6 elevation after LPS application compared to male pigs [47].

## Figures and Tables

**Figure 1 ijms-25-09283-f001:**
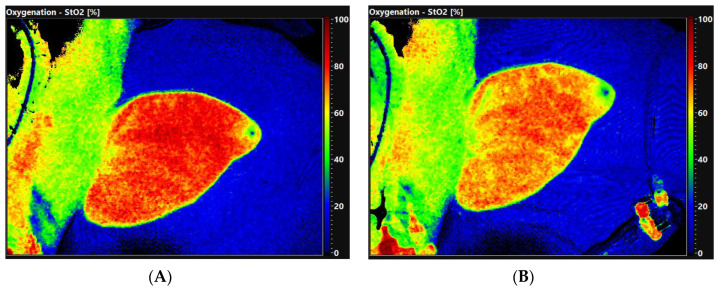
Example representation of hyperspectral imaging (HSI) measurements for tissue oxygenation (StO_2_). (**A**): H0, (**B**): H2. Red/yellow areas indicate high values (50–100), green/blue areas indicate low values (0–50).

**Figure 2 ijms-25-09283-f002:**
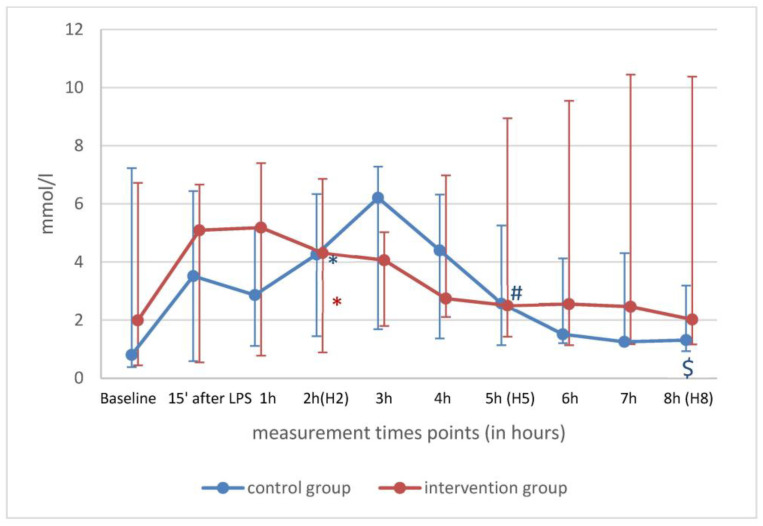
Lactate of control (blue) and intervention (red) group shown as median with minimum and maximum values over time. * *p* < 0.05 H0 vs. H2; ^#^ *p* < 0.05 H2 vs. H5; ^$^ *p* < 0.05 H2 vs. H8.

**Figure 3 ijms-25-09283-f003:**
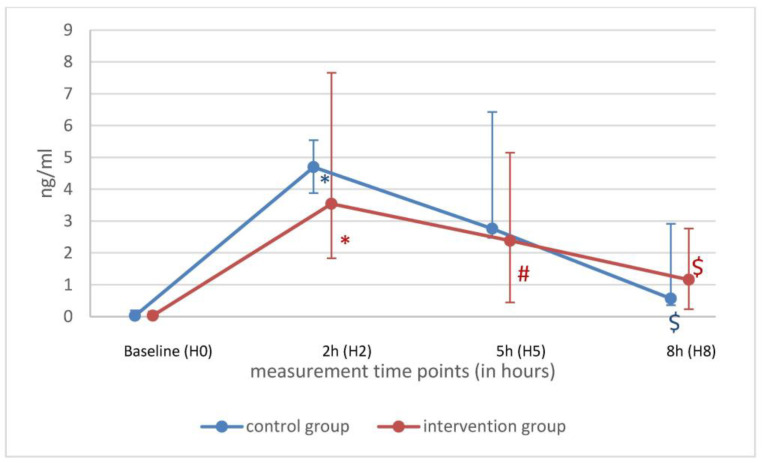
IL-6 of control (blue) and intervention (red) group shown as median with minimum and maximum values over time. * *p* < 0.05 H0 vs. H2; ^#^ *p* < 0.05 H2 vs. H5; ^$^ *p* < 0.05 H2 vs. H8.

**Figure 4 ijms-25-09283-f004:**
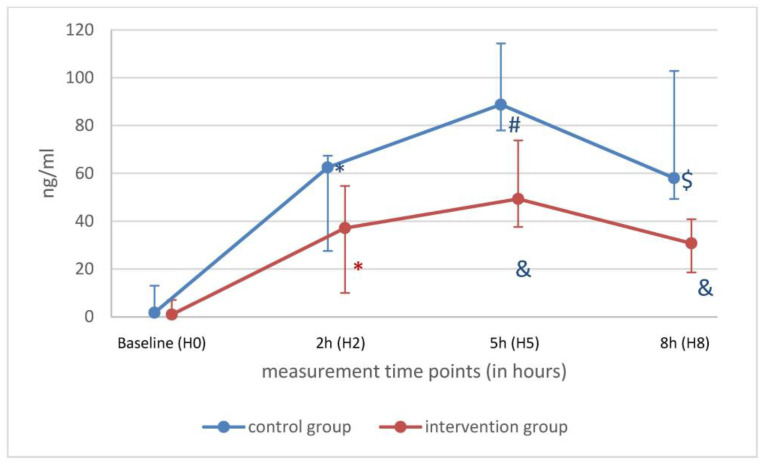
IL-1ra of control (blue) and intervention (red) group shown as median with minimum and maximum values over time. * *p* < 0.05 H0 vs. H2; ^#^ *p* < 0.05 H2 vs. H5; ^$^ *p* < 0.05 H2 vs. H8; ^&^ control vs. intervention.

**Figure 5 ijms-25-09283-f005:**
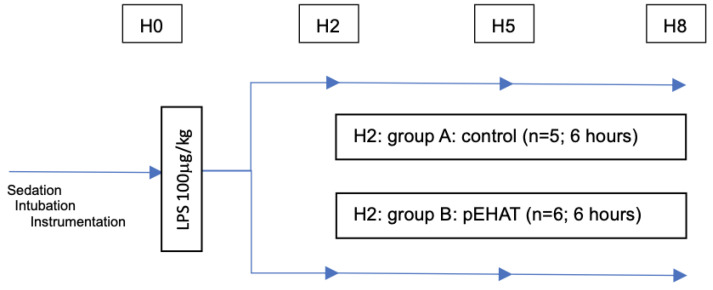
Research diagram. H0: baseline, H2: two hours after LPS infusion, H5: five hours after LPS infusion, H8: eight hours after LPS infusion. Group A: without pEHAT as the control group. Group B: pEHAT [11] with Oxiris^®^ as the intervention group.

**Table 1 ijms-25-09283-t001:** Different measurements shown as median (minimum; maximum) at baseline and 2 h after LPS infusion (H2). HR = heart rate. MAP = mean arterial pressure. PAP = pulmonal arterial pressure. CI = cardiac index. mPAP = mean pulmonary arterial pressure. p_a_O_2_ = partial arterial pressure of oxygen. p_a_CO_2_ = partial arterial pressure of carbon dioxide; VO_2_ = consumption of oxygen. VCO_2_ = consumption of carbon dioxide. IL = interleukin. TNF-α = tumor necrosis factor alpha. S_t_O_2_ = tissue oxygenation. NIR = near infrared perfusion index. TWI = tissue water index. * *p* < 0.05 H0 vs. H2 (T-Test) ^#^ *p* < 0.05 H0 vs. H2 (Wilcoxon test).

	H0 (n = 11)	H2 (n = 11)
HR (bpm)	76 (58; 133)	102 (75; 152) *
MAP (mmHg)	85 (71; 101)	85 (71; 109)
CI (l/min/m^2^)	7.1 (4.8; 10.7)	8.1 (4.9; 9.9)
Lactate (mmol/l)	1.56 (0.42; 4.38)	4.26 (0.89; 6.86) *
mPAP (mmHg)	23 (19; 38)	30 (23; 39) *
p_a_O_2_ (mmHg)	186.5 (128.8; 210.6)	173.1 (89.5; 195.0)
p_a_CO_2_ (mmHg)	48.1 (41.0; 51.2)	49.9 (42.5; 57.0)
VO_2_/m^2^ (ml/min/m^2^)	220 (187; 389)	266 (237; 374) ^#^
VCO_2_/m^2^ (ml/min/m^2^)	210 (149; 334)	272 (207; 352) *
S_t_O_2_ (%)	74.0 (63.6; 79.6)	70.4 (48.9; 75.9) *
NIR (%)	43.5 (39.7; 53.7)	45.9 (40.2; 54.8)
TWI (%)	55.5 (44.6; 58.9)	54.9 (43.7; 58.5)
IL-1β (ng/mL)	0.03 (0.00; 0.29)	0.13 (0.06; 0.67) ^#^
IL-1ra (ng/mL)	1.20 (0.19; 13.06)	53.06 (9.96; 67.36) ^#^
IL-2 (ng/mL)	0.06 (0.00; 0.84)	0.09 (0.01; 0.89) ^#^
IL-4 (ng/mL)	0.15 (0.00; 1.83)	0.24 (0.00; 1.76)
IL-6 (ng/mL)	0.03 (0.00; 0.20)	3.89 (1.83; 7.60) ^#^
IL-8 (ng/mL)	0.00 (0.00; 0.00)	0.20 (0.03; 2.75) ^#^
IL-10 (ng/mL)	0.24 (0.01; 1.58)	0.54 (0.21; 1.76) ^#^
IL-12 (ng/mL)	0.96 (0.41; 1.47)	1.16 (0.42; 2.43) ^#^
IL-18 (ng/mL)	0.20 (0.05; 3.15)	0.27 (0.08; 3.27)
TNF-α (ng/mL)	0.00 (0.00; 0.24)	1.75 (0.04; 11.00) ^#^

**Table 2 ijms-25-09283-t002:** Different measurements shown as median (minimum; maximum) at H2, H5 and H8. HR = heart rate. MAP = mean arterial pressure. PAP = pulmonal arterial pressure. CI = cardiac index. mPAP = mean pulmonary arterial pressure. p_a_O_2_ = partial arterial pressure of oxygen. p_a_CO_2_ = partial arterial pressure of carbon dioxide. VO_2_ = consumption of oxygen. VCO_2_ = consumption of carbon dioxide. S_t_O_2_ = tissue oxygenation. NIR = near infrared perfusion index. TWI = tissue water index.

		H2	H5	H8
HR (bpm)	A	120 (94; 152)	114 (77; 118)	110 (81; 134)
	B	93 (75; 128)	113 (68; 126)	95 (79; 116)
MAP (mmHg)	A	82 (71; 85)	95 (87; 116) ^#^	85 (66; 122)
	B	98 (78; 109)	96 (87; 115)	87 (80; 113)
CI (L/min/m^2^)	A	8.1 (6.5; 9.9)	5.9 (5.2; 6) ^#^	6.4 (5.7; 7)
	B	7.6 (4.9; 8.3)	6.0 (4.3; 7.9)	5.4 (5; 7.1)
Lactate (mmol/L)	A	4.3 (1.5; 6.4)	2.6 (1.1; 5.3) ^#^	1.3 (0.9; 3.2) ^$^
	B	4.3 (0.9; 6.9)	2.5 (1.4; 8.9)	2.0 (1.2; 10.4)
mPAP (mmHg)	A	26 (23; 32)	30 (20; 34) ^#^	30 (19; 33) ^$^
	B	34 (27; 39)	31 (25; 40)	30 (27; 36)
p_a_O_2_ (mmHg)	A	159.9 (134.6; 179.6)	134.0 (99.9; 164.0) ^#^	138.3 (93.8; 173.2) ^$^
	B	179.1 (89.5; 195.0)	154.9 (97.2; 178.7)	157.3 (109.0; 179.1)
p_a_CO_2_ (mmHg)	A	51.0 (45.1; 57.0)	44.1 (40.4; 54.2)	43.6 (39.9; 48.3)
	B	46.3 (42.5; 53.5)	46.8 (38.3; 59.9)	45.1 (38.5; 55.1)
VO_2_/m^2^ (mL/min/m^2^)	A	287 (237; 374)	296 (200; 355)	282 (205; 346)
	B	260 (244; 365)	247 (190; 317)	240 (207; 284)
VCO_2_/m^2^ (mL/min/m^2^)	A	273 (207; 352)	262 (183; 311)	275 (180; 286)
	B	264 (227; 311)	231 (192; 297) ^#^	232 (202; 280) ^$^
S_t_O_2_ (%)	A	68.5 (59.8; 73.3)	65.4 (60.4; 69.9)	66.6 (58.3; 72.9)
	B	71.3 (48.9; 75.9)	68.1 (59.4; 76.1)	66.3 (50.3; 73.8)
NIR (%)	A	45.9 (41.5; 54.8)	40.2 (32.2; 48.9)	41.6 (34.5; 50.2)
	B	43.8 (40.2; 49.3)	44.6 (34.6; 51.5)	40.2 (36.7; 48.8)
TWI (%)	A	55.5 (52.8; 57.7)	60.8 (53.1; 62.2)	61.3 (52.4; 63.4)
	B	50.1 (43.7; 58.5)	49.8 (45.1; 63.9)	53.1 (48.9; 66.2)

^#^ *p* < 0.05 H2 vs. H5; ^$^ *p* < 0.05 H2 vs. H8.

**Table 3 ijms-25-09283-t003:** Different measurements shown as median (minimum; maximum) at H2, H5 and H8. IFN = interferon. IL = interleukin. TNF-α =tumor necrosis factor alpha. * *p* < 0.05 group A vs. B; ^#^ *p* < 0.05 H2 vs. H5; ^$^ *p* < 0.05 H2 vs. H8.

		H2	H5	H8
IFN-γ (pg/mL)	A	0.00 (0.00; 0.09)	0.00(0.00; 0.28)	0.00 (0.00; 0.00)
	B	0.05 (0.00; 0.61)	0.14 (0.00; 0.61)	0.05 (0.00; 0.42)
IL-1α (pg/mL)	A	0.02 (0.00; 0.04)	0.02 (0.00; 0.05)	0.02 (0.00; 0.04)
	B	0.03 (0.02; 0.14)	0.03 (0.01; 0.16)	0.03 (0.01; 0.13)
IL-1β (pg/mL)	A	0.10 (0.06; 0.33)	0.29 (0.13; 1.23) ^#^	0.23 (0.07; 1.00) ^$^
	B	0.14 (0.06; 0.67)	0.29 (0.15; 2.07)	0.23 (0.13; 1.39)
IL-1ra (pg/mL)	A	62.50(27.56; 67.36)	88.78 (77.98; 114.33) ^#^	58.03 (49.34; 102.87)
	B	37.14 (9.96; 54.77)	49.33 (37.61; 73.79) *	30.77 (18.58; 40.87) *
IL-2 (pg/mL)	A	0.05 (0.00; 0.12)	0.05 (0.01; 0.14)	0.03 (0.00; 0.10)
	B	0.14 (0.02; 0.89)	0.15 (0.01; 0.86)	0.15 (0.01; 0.79)
IL-4 (pg/mL)	A	0.15 (0.00; 0.49)	0.22 (0.03; 0.67)	0.19 (0.00; 0.42)
	B	0.33 (0.00; 1.76)	0.28(0.00; 2.04)	0.33 (0.00; 1.81)
IL-6 (pg/mL)	A	4.70 (3.88; 5.55)	2.76 (2.50; 6.43)	0.568 (0.359; 2.916) ^$^
	B	3.54 (1.83; 7.66)	2.39 (0.45; 5.15) ^#^	1.16 (0.24; 2.77) ^$^
IL-8 (pg/mL)	A	0.25 (0.13; 0.66)	0.00 (0.000; 0.02) ^#^	0.00 (0.00; 0.00) ^$^
	B	0.16 (0.03; 2.75)	0.00 (0.000; 0.04) ^#^	0.00 (0.000; 0.01) ^$*^
IL-10 (pg/mL)	A	0.54 (0.44; 0.89)	0.36 (0.26 0.95)	0.20 (0.14; 0.53)
	B	0.57 (0.21; 1.76)	0.42 (0.10; 1.58) ^#^	0.45 (0.09; 1.44) ^$^
IL-12 (pg/mL)	A	1.36 (0.83; 2.43)	1.52 (0.63; 3.86)	1.44 (0.51; 3.24)
	B	1.02 (0.42; 1.83)	0.80 (0.44; 1.43) ^#^	0.63 (0.34; 1.19) ^$^
IL-18 (pg/mL)	A	0.27 (0.22; 0.60)	0.35 (0.23; 0.97)	0.32 (0.20; 1.03)
	B	0.34 (0.08; 3.27)	0.54 (0.12; 3.56)	0.72 (0.18; 3.71)
TNF-α (pg/mL)	A	2.57 (0.51; 11.00)	0.26 (0.00; 0.58) ^#^	0.10 (0.00; 0.18) ^$^
	B	1.15 (0.04; 2.70)	0.03 (0.00; 0.09) ^#^	0.02 (0.00; 0.04) ^$^

## Data Availability

Resource data are available from the corresponding author on reasonable request.

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
