# Peer review of "Effects of Oxiris® Therapy on Cytokine Elimination after a LPS Infusion—An Experimental Animal Study"

_ijms, 2024, doi:10.3390/ijms25179283_

Round 1

Reviewer 1 Report (New Reviewer)

Comments and Suggestions for Authors

In their study, Kalenka et al. used a large animal model of endotoxic shock to show the relevance of the Oxiris® filter to remove LPS and pro- as well as anti-inflammatory cytokines. A single high bolus of 100 µg/kg bw LPS was intravenously applied for 2 h to 11pigs. Afterwards 6 pigs were treated with a Oxiris® filter system via pEHAT and 5 pigs remained untreated as control group for further 6 h. Animals were monitored at timepoints 0, 2, 5, and 8 h. This means that timepoints 5 h and 8 h were 3 h and 6 h after setting up the Oxiris® filter system or without as control group. The authors observed in line to their expectation an increase in pro- and anti-inflammatory cytokines as well as a sepsis-like cardiocirculatory parameters such as tachycardia, elevated pulmonary pressure, and increase lactate levels upon LPS challenge. Interestingly they did not find a reduced cytokine level in the Oxiris® filter system treated animals compared to the control group at the 5 h and 8 h timepoints, although in both groups at both timepoints pro-inflammatory cytokine were significantly reduced.

The study is of interest, because the effective treatment of sepsis is still a general need. However, in my opinion, the setup of the study should be discussed in relationship to already performed studies with a similar setup. At least for me, it is not clear how the used setup can be compared with already performed studies and what is the conclusion. Therefore, the discussion should be improved.

Some minor concerns should also be addressed:

Line 65: Female vs male; did the author have an assumption whether the experiments when performed with male pigs have a similar outcome?

Is the age of the animals important? It is not mentioned.

Lines 94-99: Figure 1 is not correctly shown. Please correct.

Lines 27, 126, 153, 168, Legend Table 1 and Table 1, 175, 178, 208, Table 3, 235: TNF α should be TNF-α

Lines 153, 178, 292:  Il should be IL

Lines 41, 52, 221, 228, 229, 251, 256/257, 284, 303/304: References should ben combined.

Lines 91/92: Unclear. Please rewrite.

Lines 155, 160,  Legend Table 1 and Table 1,  Table 2, 299: StO2 should be StO2

Lines 262, 267, 289, 302: ml or mL

Line 303: IL-1Ra should be IL-1ra

Author Response

Reviewer No.1:

We would like to thank the reviewer for the comments. They surely will improve our paper.

Comment 1:

The study is of interest, because the effective treatment of sepsis is still a general need. However, in my opinion, the setup of the study should be discussed in relationship to already performed studies with a similar setup. At least for me, it is not clear how the used setup can be compared with already performed studies and what is the conclusion. Therefore, the discussion should be improved.

Answer 1:

We would like to thank the reviewer for the suggestions. We realize that our explanation may not have been precise enough. Our aim was not to establish a sepsis model, but rather to conduct an inflammation model. We have already attempted to clearly address this distinction.

We are not aware of performed studies with a setup similar to ours. Nonetheless we included now a study in humans with cytosorb therapy after a LPS Challenge. We included a study in pigs with a smoke and burn induced inflammation. In addition, we discussing relevant studies in the context of infection and sepsis with elevated cytokines.

First paragraph of the discussion: “We chose an established model for the controlled simulation of an inflammatory stimulus in pigs. Due to its potency as an immune system stimulator, lipopolysaccharide has been extensively used across various animal species to investigate gram-negative bacterial infections and sepsis. Experimental endotoxemia primarily represents a systemic challenge without an initial infectious focus and lacks the characteristic immune response induced by sepsis [17]. LPS injection is widely accepted as a model for studying the hypodynamic phase of Gram-negative bacterial shock, rather than sepsis [18]. We chose a high single bolus dose of LPS (100 µg/kg bw). Usually, doses ranging from 0.5 to 75 µg/kg bw are used [19]. We selected this dose to generate a relevant IL-6 response (4 pg/ml) as the primary marker of inflammatory response to the LPS stimulus [20] [21] [22]. However, using the high dose allowed us to achieve the desired cardio-pulmonary and inflammatory effect. Nevertheless, the administration of vasopressors was not necessary at any time. Higher LPS bolus doses up to 150 µg/kg bw produce an experimental model for endotoxemic shock in pigs [23].”

Starting line 277: “Studies have confirmed that elevated endotoxin levels in the body can significantly increase patient mortality [34]. As a result, the prompt and effective removal of endotoxins and inflammatory mediators, may be a crucial treatment option. The current evidence supporting the use of extracorporeal hemadsorption technique is not conclusive. However, available data indicate that these techniques are safe and most effective during the early phase of septic shock, particularly within the first 24 hours of the dysregulated host response [35] [36].”

New: “A human study involving 24 volunteers who were challenged with an LPS infusion showed that in subjects of the extracorporeal hemadsorption (CytoSorb®) group, the increase in plasma cytokine concentrations after the first LPS challenge was significantly attenuated compared to the control group. In this study, hemadsorption was initiated as early as 15 minutes after the initial LPS bolus.”

New: ”An animal study in a porcine model of smoke and burn injury, hemadsorption with the CytoSorb® system did not lead to significant reductions in the measured cytokines”

Minor comments:

Comment 2:

Female vs male; did the author have an assumption whether the experiments when performed with male pigs have a similar outcome?

Answer 2:

We modified this now in the discussion: “Sex had a significant impact on the immune responses after LPS challenge. Therefore, we used only female pigs, as they exhibit greater TNF-α and IL-6 elevation after LPS administration compared to male pigs.”

Comment 3:

Figure 1 is not correctly shown

Answer 3:

We realized that the figure is not correctly shown most probably because of the editioring process or transfer to a pdf. We tried to do it better.

Several comments:

Several typos and reduction of the literature references have been implemented.

Reviewer 2 Report (New Reviewer)

Comments and Suggestions for Authors

Major Concern:

The study evaluates the effectiveness of Oxiris in reducing cytokines using an animal model, but it does not show significant effects. Additionally, the research suggests the potential efficacy of pEHAT; however, it lacks a control experiment. The drawn conclusions appear constrained due to these limitations. I would suggest the authors reconsider the framing of their main conclusions to enhance the significance of the findings. One approach could be to restructure the conclusions in a manner that emphasizes the innovative use of this particular large animal model in studying significant inflammatory responses.

Minor Concerns:

Introduction: It would be beneficial if the authors could provide a brief overview of existing animal models used in sepsis treatment studies. Additionally, clarification on the choice of animal model for Oxiris testing would enhance the reader's understanding.

Lines 90-92: The manuscript mentions cytokine level assessments at 2hr, 5hr, 8hr, and further 6 hr intervals, yet Figure 4 shows measurements taken every 1 hour for both groups. Could the authors explain for this discrepancy?

Figure 1: The research diagram in Figure 1 requires adjustments for clearer label representation, ensuring that readers can easily understand the content of this figure.

Line 153: The formatting of interleukin abbreviations needs standardization (e.g., converting "Il-6, Il-8" to "IL-6, IL-8"). Similarly, the representation of "TNF α" should be corrected to "TNF-α" throughout the text for consistency.

Figure 4: the error bar of the intervention group is rather high, especially after 5 hours, suggesting high variability. Could the author clarify the data analysis method. Whether the data represent median values across multiple groups? How the conclusions were drawn under these conditions?

In the discussion, the authors mentioned "established model" and "various animal species" without justifying the current model's selection. Aligning with the first minor concern, a detailed rationale for this choice, including a comparison with other tested models (if applicable) would be better.

Comments on the Quality of English Language

The English Language in this manuscript is well organized and easy to understand.

Author Response

Reviewer No. 2:

Major Concern:

Comment 1:

The study evaluates the effectiveness of Oxiris in reducing cytokines using an animal model, but it does not show significant effects. Additionally, the research suggests the potential efficacy of pEHAT; however, it lacks a control experiment. The drawn conclusions appear constrained due to these limitations. I would suggest the authors reconsider the framing of their main conclusions to enhance the significance of the findings. One approach could be to restructure the conclusions in a manner that emphasizes the innovative use of this particular large animal model in studying significant inflammatory responses.

Answer 1:

We thank the reviewer for the recommendations. We believe that the group without hemadsorption provides a meaningful control group. Nevertheless, we have revised the conclusions to moderate their implications and soften their impact.

Abstract and conclusion new “The presented large animal model may be a promising option for studying the effects of hemadsorption techniques.”

Minor Concerns:

Comment 2:

Introduction: It would be beneficial if the authors could provide a brief overview of existing animal models used in sepsis treatment studies. Additionally, clarification on the choice of animal model for Oxiris testing would enhance the reader's understanding.

Answer 2:

We would like to thank the reviewer. Our intention was not to establish a sepsis model, but rather to use an inflammatory model. We understand that we were not precise enough in our wording. To clarify this, we have now started the discussion with the sentence " We selected a large animal pig model because porcine anatomy and physiology are generally quite similar to those of humans [15], making the transfer of knowledge more applicable. Especially blood flow over the hemadsorption device is in great analogy to humans [11]. For a more detailed insight into inflammatory large animal models, we refer to a recent review [16].

Experimental endotoxemia primarily represents a systemic challenge without an initial infectious focus and lacks the characteristic immune response induced by sepsis [16]. We chose an established model for the controlled simulation of an inflammatory stimulus in pigs [17]. "

Comment 3:

Lines 90-92: The manuscript mentions cytokine level assessments at 2hr, 5hr, 8hr, and further 6 hr intervals, yet Figure 4 shows measurements taken every 1 hour for both groups. Could the authors explain for this discrepancy?

Answer 3:

Indeed this is confusing. We measured vital parameter and blood gases every hour. For the sake of simplicity we only presented data in the tables at mentioned time points. Cytokine were only measured at mentioned time points.

We clarified this now: “After the initial instrumentation, baseline measurements (heart rate (HR), mean arterial pressure (MAP), pulmonary artery pressure (PAP), cardiac index (CI), blood gases, VO2 and VCO2 and hyperspectral imaging) were measured. An intravenous dose of LPS (100 µg/kg bw O111:B4, LPS-EB, InvivoGen, Toulouse, France) was then administered over 15 minutes. Measurements were recorded immediately after LPS administration and every hour following the baseline measurements. For the sake of simplicity these data are presented in the tables at baseline (H0), 2 hours (H2), 5 hours (H5), and 8 hours (H8) only. Cytokine levels were collected at baseline (H0), 2 hours (H2), 5 hours (H5), and 8 hours (H8).”

Comment 4:

Figure 1: The research diagram in Figure 1 requires adjustments for clearer label representation, ensuring that readers can easily understand the content of this figure.

Answer 4:

Thank you for the advice. We label now the diagram more clearer.

Comment 5:

Line 153: The formatting of interleukin abbreviations needs standardization (e.g., converting "Il-6, Il-8" to "IL-6, IL-8"). Similarly, the representation of "TNF α" should be corrected to "TNF-α" throughout the text for consistency.

Answer 5:

Thanks. Done

Comment 5:

Figure 4: the error bar of the intervention group is rather high, especially after 5 hours, suggesting high variability. Could the author clarify the data analysis method. Whether the data represent median values across multiple groups? How the conclusions were drawn under these conditions?

We thank the reviewer for the attentive interpretation. The presented data are shown as minimum and maximum values (as described in the legend). In our experience, such variability is not unusual in a large animal pig experiment. The statistical analyses were conducted following testing normal distribution with the tests recommended by our medical statistician. As already mentioned in the limitations, we believe that the small group size must be taken into account in the data interpretation.

Comment 6:

In the discussion, the authors mentioned "established model" and "various animal species" without justifying the current model's selection. Aligning with the first minor concern, a detailed rationale for this choice, including a comparison with other tested models (if applicable) would be better.

Answer 6:

Thank you for these suggestions.

New start of discussion: “We selected a large animal pig model because porcine anatomy and physiology are generally quite similar to those of humans [15], making the transfer of knowledge more applicable. Especially the blood flow through the hemadsorption device closely resembles that in humans. [11]. Experimental endotoxemia primarily represents a systemic challenge without an initial infectious focus and lacks the characteristic immune response induced by sepsis [16]. We chose an established model for the controlled simulation of an inflammatory stimulus in pigs [17].”

Round 2

Reviewer 2 Report (New Reviewer)

Comments and Suggestions for Authors

Thank the authors for clarifying. My raised concern has been addressed.

Comments on the Quality of English Language

The quality of English language is clear and easy to understand.

This manuscript is a resubmission of an earlier submission. The following is a list of the peer review reports and author responses from that submission.

Round 1

Reviewer 1 Report

Comments and Suggestions for Authors

The authors explain the choice of a high single dose of LPS with the desire to obtain the desired effect, but do not mention the potential negative consequences of such a dose, which may not reflect the actual inflammation.

Why does the abstract describe a detailed research plan (who, what, how much)? Such information is not likely to be provided here, but in the Materials and Methods (or similar) chapter.

The control group (5 pieces) and the intervention group (6 pieces) are probably not enough for the analysis results to be considered relatively reliable.

The article lacks a research diagram (preferably in graphical form).

Table 2 is formatted very unclearly (it goes on for 3 pages), some columns are too narrow and the text wraps.

As you know, Wilcoxon and Man Whitney-U tests should only be used when it is not possible to use parametric tests (here: t-test in two variants: for repeated and unrepeated measures). In the article, the authors do not write anything about checking the normality conditions. When normality tests are positive, the t-test should be used instead of a parametric one. Please clarify this issue.

Author Response

Reviewer 1:

Comment 1:

The authors explain the choice of a high single dose of LPS with the desire to obtain the desired effect, but do not mention the potential negative consequences of such a dose, which may not reflect the actual inflammation.

Answer comment 1:

We would like to thank the reviewer for this suggestion. Indeed, we have somewhat focused on the desired effects of an LPS infusion. We have now included a more detailed description of the effects of an LPS infusion in the discussion.

“We chose an established model for the controlled simulation of an inflammatory stimulus in pigs. Due to its potency as an immune system stimulator, lipopolysaccharide has been extensively used across various animal species to investigate gram-negative bacterial infections and sepsis. Experimental endotoxemia primarily represents a systemic challenge without an initial infectious focus and lacks the characteristic immune response induced by sepsis {Poli-de-Figueiredo, 2008 #101}. LPS injection is widely accepted as a model for studying the hypodynamic phase of Gram-negative bacterial shock, rather than sepsis {Fink, 1990 #102}.

Comment 2:

Why does the abstract describe a detailed research plan (who, what, how much)? Such information is not likely to be provided here, but in the Materials and Methods (or similar) chapter.

Answer comment 2:

Indeed the abstract is too long and detailed. We shorten the abstract but still try to give interesting and maybe necessary information in the abstract.

Comment 3:

The control group (5 pieces) and the intervention group (6 pieces) are probably not enough for the analysis results to be considered relatively reliable.

Answer comment 3:

Thanks for that advice. We now include this in the limitation section.

“The relative small number of animals per group are probably not sufficient for the analysis results to be considered relatively reliable.

Comment 4:

The article lacks a research diagram (preferably in graphical form).

Answer comment 4:

Please find a research diagram now as figure 1.

Comment[i] 5:

Table 2 is formatted very unclearly (it goes on for 3 pages), some columns are too narrow and the text wraps.

Answer comment 5:

Thanks for that comment. During the editoring process the columns change their format. We now include 2 new tables instead of one table for these data. We clarify significance now by symbols and not by separate columns.

Comment 6:

As you know, Wilcoxon and Man Whitney-U tests should only be used when it is not possible to use parametric tests (here: t-test in two variants: for repeated and unrepeated measures). In the article, the authors do not write anything about checking the normality conditions. When normality tests are positive, the t-test should be used instead of a parametric one. Please clarify this issue.

Answer comment 6:

We reanalyzed our data.

Please find now “Baseline values were tested for normal distribution. In the case of a normal distribution, a paired t-test was performed; otherwise, the Wilcoxon test was used to compare values in between the group at timepoint H0 and H2 (SPSS, IBM, Version 28.0). Data between groups were analyzed by Man Whitney-U test. Values are shown as median (minimum; maximum) values. A p-value of less than 5% (p < 0.05) was considered statistically significant.”

Reviewer 2 Report

Comments and Suggestions for Authors

Dear authors, please find my comment. 

Abstract

Please specify the critical findings in the abstract, particularly the impact of Oxiris® on cytokine levels.

Introduction

In the introduction, please provide more explanation about sepsis and describe the mechanism better.

Is the rationale for using Oxiris® therapy in sepsis management adequately justified concerning existing literature?

Materials and Methods

Why were female domestic pigs chosen explicitly for this study?

Could you provide more detail on the selection criteria and the generalizability of this model to human sepsis?

The methods mention randomising animals into treatment and control groups. How was randomisation carried out to ensure unbiased group assignments?

The study uses a high dose of LPS (100 µg/kg bw). Can the authors explain why this dose was chosen over others and how it compares to similar studies? Add an explanation in the Discussion section.

Are the methods for cytokine measurement (Luminex-based multiplex assay) sufficiently detailed?

Are there any validation steps or controls to ensure the accuracy and reproducibility of these measurements?

Elaborate on why specific statistic tests were chosen and whether adjustments for multiple comparisons were considered.

Results

The results indicate no significant differences in cytokine levels between the treatment and control groups. Could the authors provide more context or potential reasons for these findings?

Were any statistically significant differences found that might suggest subtle effects of the Oxiris® filter not immediately apparent from the primary outcomes?

Discussion

Please provide a balanced interpretation of your findings, considering their study's strengths and limitations.

How do these findings align or contrast with previous studies on hemadsorption and cytokine elimination in sepsis models? Are there any critical studies that should be discussed to provide context?

The discussion mentions the potential clinical implications of the findings. How might the results influence clinical practice or future research directions?

The summary needs to summarise the main findings and their significance briefly.

Could you outline straightforward suggestions for future research? What aspects should you consider next to address the gaps identified in this study?

Author Response

Reviewer Nr. 2.

We would like to thank the reviewer for the comments. These will surely improve our paper. Nonetheless, we are a bit confused about which version reviewer No. 2 had for their review. We submitted the paper on June 26th and received two reviewer comments on July 16th. One review was obviously wrongly attached to our paper. We resubmitted our paper 21st July answering the comments of reviewer 1. The comments of the “new” reviewer 2 appeared 23rd July. The editorial office could not help us determine which version reviewer 2 had. We assume that reviewer 2 had the initial submission. Some changes were already made with the resubmission. We hope to resolve the discrepancies.

Comment 1:

Abstract

Please specify the critical findings in the abstract, particularly the impact of Oxiris® on cytokine levels.

Answer to comment 1: We would like the reviewer for that suggestion. The abstract in the actual version is 200 words long. This is exact the wordcount which is recommended in the guidelines for authors.

In addition, we could not find any relevant impact of the Oxiris® filter on cytokine levels. Therefore, we think it would be excessive to include non-significant differences in cytokine levels in the abstract.

“No differences between the intervention and the control group could be detected after 3 and 6 hours for IL-1β, IL-2, IL-6, IL-8, IL-12 and TNF α, suggesting no effect of the Oxiris® filter on the elimination of elevated cytokines. We were unable to demonstrate any detectable effects of the Oxiris® filter on the elevated cytokines following a lipopolysaccharide infusion in this large animal model”

Comment 2

Introduction

In the introduction, please provide more explanation about sepsis and describe the mechanism better.

Answer to comment 2: We would like to highly appreciate this comment. It looks like we were not precise enough. We were not aiming for a sepsis model. Our goal was an inflammation model. Nonetheless, Oxiris® is mainly use in sepsis.

We applied following modifications:

Abstract: we deleted “in the context of sepsis” as this might mislead the readers.

Introduction: we included now “Sepsis is a life-threatening condition, most commonly triggered by bacterial and other infections, characterized by the host's dysregulated inflammatory response, which involves both proinflammatory cytokines and a compensatory anti-inflammatory response.”

Discussion: we include now “Due to its potency as an immune system stimulator, lipopolysaccharide has been extensively used across various animal species to investigate gram-negative bacterial infections and sepsis. Experimental endotoxemia primarily represents a systemic challenge without an initial infectious focus and lacks the characteristic immune response induced by sepsis [17]. LPS injection is widely accepted as a model for studying the hypodynamic phase of Gram-negative bacterial shock, rather than sepsis [18].”

Comment 3:

Is the rationale for using Oxiris® therapy in sepsis management adequately justified concerning existing literature?

Answer to comment 3:

Thank you for this excellent question. As mentioned before we did not do a sepsis model. In the introduction we mentioned “Blood purification therapies are undergoing significant advancements and gaining traction in sepsis management, propelled by both technological enhancements and the steadfast belief among a segment of the medical community in their efficacy for septic shock [6].” We think there is “only” belief. This is what we pointed out.

Comment 4:

Why were female domestic pigs chosen explicitly for this study?

Answer to comment 4: There was “no specific reason”.

Comment 5:

Could you provide more detail on the selection criteria and the generalizability of this model to human sepsis?

Answer to comment 5:

We included in the revised manuscript already:

Discussion: “Experimental endotoxemia primarily represents a systemic challenge without an initial infectious focus and lacks the characteristic immune response induced by sepsis [17]. LPS injection is widely accepted as a model for studying the hypodynamic phase of Gram-negative bacterial shock, rather than sepsis [18].”

Limitation paragraph: “This study has several limitations. First, this is a large animal model and therefore the results cannot be applied to human patients without any restrictions. Secondly, the relative small number of animals per group are probably not sufficient for the analysis results to be considered relatively reliable.”

Comment 6:

The methods mention randomising animals into treatment and control groups. How was randomisation carried out to ensure unbiased group assignments?

Answer to comment 6: in M&M: “The animals were assigned to the two groups using a computer-generated randomizer. treatment 120 minutes after LPS infusion (H2) and monitored for further 6 hours (figure 1).” Also included now a research diagram.

Comment 7:

The study uses a high dose of LPS (100 µg/kg bw). Can the authors explain why this dose was chosen over others and how it compares to similar studies? Add an explanation in the Discussion section.

Answer comment 7:

In the revised manuscript is already mentioned: “Experimental endotoxemia primarily represents a systemic challenge without an initial infectious focus and lacks the characteristic immune response induced by sepsis [17]. LPS injection is widely accepted as a model for studying the hypodynamic phase of Gram-negative bacterial shock, rather than sepsis [18]. We chose a high single bolus dose of LPS (100 µg/kg bw). Usually, doses ranging from 0.5 to 75 µg/kg bw are used [19].”

We now extended this paragraph: “We selected this dose to generate a relevant IL-6 response (4 pg/ml) as the primary marker of inflammatory response to the LPS stimulus [20] [21] [22]. However, using the high dose allowed us to achieve the desired cardio-pulmonary and inflammatory effect. Nevertheless, the administration of vasopressors was not necessary at any time. Higher LPS bolus doses up to 150 µg/kg bw produce an experimental model for endotoxemic shock in pigs [23].”

Comment 8

Are the methods for cytokine measurement (Luminex-based multiplex assay) sufficiently detailed?

Answer comment 8:

We would like to thank the reviewer for this comment. We considered extending this paragraph but believe that more details are very specialized and do not provide relevant information for the majority of readers.

Comment 9:

Are there any validation steps or controls to ensure the accuracy and reproducibility of these measurements?

Answer comment 9:

One of our co-authors is an expert in the field of laboratory medicine (MB). He took especially take care of high-end standards for the cytokine analyses. We extended this. Paragraph: “Samples were run in triplicate against internal controls, and the mean value was used for further statistical analysis. We estimated an elimination half-time for IL-6 and TNF α of three hours based on available cytokine kinetics from porcine model [15] [16].

Comment 10:

Elaborate on why specific statistic tests were chosen and whether adjustments for multiple comparisons were considered.

Answer comment 10:

Please find in the revised version “Baseline values were tested for normal distribution. In the case of a normal distribution, a paired t-test was performed; otherwise, the Wilcoxon test was used to compare values in between the group at timepoint H0 and H2 (SPSS, IBM, Version 28.0).”

After consultation of an expert in medical statistics at Heidelberg University we did no multiple comparisons because of the relative small numbers per group.

Comment 11

Results

The results indicate no significant differences in cytokine levels between the treatment and control groups. Could the authors provide more context or potential reasons for these findings?

Answer comment 11:

We appreciate this complex question. As we already attempt to describe in the discussion section, we aim to place our data in context with the existing literature. Essentially, there are no differences, perhaps indicating no effect. In a deeper context, several studies that claimed to show an effect have various limitations, which we discuss in the discussion section. Due to the relatively small number of animals per group, we prefer not to speculate further on potential reasons.

Comment 12

Were any statistically significant differences found that might suggest subtle effects of the Oxiris® filter not immediately apparent from the primary outcomes?

Answer to comment 12:

We established a stringent animal model. To our knowledge, we conducted an extensive data analysis, including micro- and macrovascular monitoring every hour, as well as VO2 and VCO2 data. For the sake of clarity, we primarily present data at the specified time points. Nonetheless, we did not find any relevant differences at other time points.

Comment 13

Discussion

Please provide a balanced interpretation of your findings, considering their study's strengths and limitations.

Answer comment 13:

We would like to thank the reviewer. We have attempted to present a balanced interpretation of our data in the context of the existing literature. We have also consciously avoided speculative statements. We extended the strengths and limitations paragraph already

“Our study has several strengths. We utilized an uniform stimulus to induce inflammation in a simulated, large animal model. We conducted extensive analyses at both micro- and macrocirculatory levels, as well as across a broad panel of cytokine values. Using the pEHAT method, we eliminated convective elimination associated with traditional RRT.

This study has several limitations. First, this is a large animal model and therefore the results cannot be applied to human patients without any restrictions. Secondly, the relative small number of animals per group are probably not sufficient for the analysis results to be considered relatively reliable. Third, we simulated a gram-negative infection through LPS infusion. The effects of HA in a gram-positive infection may be different. Fourth, we do not have a comparison group with Oxiris® filter using the pEHAT technique and a classical RRT.”

Comment 14:

How do these findings align or contrast with previous studies on hemadsorption and cytokine elimination in sepsis models? Are there any critical studies that should be discussed to provide context?

Answer to comment 14:

We are not sure what the suggestion of the reviewer is here. We already included in the discussion paragraph:

“A human study showed that when Oxiris® was implemented in a RRT system the average blood flow rate was 132 ± 69 ml/min [29]. A cross-over regime consisting of 24 h treatment with an Oxiris® filter followed by 24 h treatment with a standard filter or the reverse filter order was used in this study. Cytokine levels significantly decreased over time in both groups. TNF α decreased more significantly with the Oxiris® filter, while IL-6 and IL-8 remained at similar levels in both groups.

In a non-randomized study RRT was conducted using either the Oxiris® or a classic AN69-ST filter with a blood flow rate between 150 and 200 mL/min. The choice of filter was determined by the patient or their relatives, since the Oxiris® filter was not covered by insurance, and it was not the preferred option for some patients. In this study several cytokines decreased in both filters after 24 h, significantly more in the Oxiris® group [37].

A retrospective observational study included 70 patients with suspected or proven gram-negative infection and acute renal failure. These patients were treated with Oxiris® filter and were matched to 66 historical patients as control. IL-6 levels were higher at the start of the study in the Oxiris® group and decreased in both groups with a greater reduction observed in the Oxiris® group [38].

Twenty-three septic patients were treated with Oxiris® in another study. The analyses included another 30 patients which were treated at the same time period, serving as controls. The levels of IL-6, IL-10 and endotoxin in both groups decreased significantly. Moreover, the Oxiris® group showed significantly lower levels of these cytokines to the control group after 3 days treatment [39].

All the mentioned four studies introduced the Oxiris® filter together with a RRT. The “peak concentration hypothesis” proposes that continuous RRT could be advantageous for the treatment of sepsis. Due to their continuous nature and non-specific removal capabilities, RRT may help reduce the peaks of both pro-inflammatory and anti-inflammatory mediator concentrations, thereby restoring immunohomeostasis [40]. Several studies have indicated that cytokine removal by RRT is primarily attributed to adsorption rather than convective transport [41] [42].”

Comment 15:

The discussion mentions the potential clinical implications of the findings. How might the results influence clinical practice or future research directions?

Answer comment 15:

We now included at the end of the discussion: “We deliberately aimed to present a clear data analysis in the context of existing evidence with our paper. From our perspective, our data support the use of hemoadsorption only within the framework of a clinical study.”

We deleted: “Given the data available, an Oxiris® therapy cannot be considered standard practice. In particular, a combination of HA and ECMO appears to result in higher mortality [47].”

Comment 16:

The summary needs to summarise the main findings and their significance briefly.

Comment 17:

Could you outline straightforward suggestions for future research? What aspects should you consider next to address the gaps identified in this study?

Answer to comment 16 and 17:

We extended the summary paragraph.

“In summary, an infusion of 100 µg/kg in this large animal model resulted in a significant inflammatory response. Both pro-inflammatory and anti-inflammatory cytokines in the plasma increased significantly following LPS administration. We implemented a pumpless extracorporeal hemadsorption technique (pEHAT), which functions as an HA method, in the intervention group for 6 hours without detectable problems. However, we were unable to demonstrate any significant effects of the Oxiris® filter on the elevated cytokines following LPS infusion in this large animal model. We aimed to present a clear data analysis within the context of existing evidence in our paper. From our perspective, our data support the use of hemoadsorption only within the framework of a clinical study. In our opinion, these studies should include coordinated patient selection and measurements of cytokine levels”

Reviewer 3 Report

Comments and Suggestions for Authors

Minor revision
The manuscript presented is intriguing and aligns well with the scope of IJMS.

    The functionality of the Oxiris filter should be clearly elucidated and introduced in the manuscript.
    In cases where dereferences in analyte levels are either significant or not significant, please explicitly state this in the corresponding graph titles. Additionally, in the tables presented, ensure to indicate any observed significant differences.
    If there are comparable results available for other animal models, it would be beneficial to discuss these findings.
    It would be beneficial to provide a more detailed introduction to the use of LPS in the study.

Minor:
The dual system for labeling experimental time (hX and HX) may cause confusion and should be replaced, or better clarified.

Author Response

Thank you for the comments. Unfortunately, we could not find any relevance to our study. Could it be possible that the review was mistakenly assigned to our study?

Round 2

Reviewer 2 Report

Comments and Suggestions for Authors

Dear authors,

Thank you for accepting all suggestions. Your manuscript is ready for publication. 

Best wishes

Author Response

We would like to thank the reviewer for the support.